

# Universality vs experience: a cross-cultural pilot study on the consonance effect in music at different altitudes

Giulia Prete[1,*], Danilo Bondi[2,*], Vittore Verratti[1], Anna Maria Aloisi[3], Prabin Rai[4,5] and Luca Tommasi[1]

[1] Department of Psychological, Health and Territorial Sciences, "G. d'Annunzio" University of Chieti-Pescara, Chieti, Italy
[2] Department of Neuroscience, Imaging and Clinical Sciences, "G. d'Annunzio" University of Chieti-Pescara, Chieti, Italy
[3] Department of Medicine, Surgery and Neuroscience, University of Siena, Siena, Italy
[4] Unique College of Medical Science and Hospital, Rajbiraj, Nepal
[5] Mechi Technical Training Academy, Birtamode, Nepal
* These authors contributed equally to this work.

## ABSTRACT

**Background:** Previous studies have shown that music preferences are influenced by cultural "rules", and some others have suggested a universal preference for some features over others.

**Methods:** We investigated cultural differences on the "consonance effect", consisting in higher pleasantness judgments for consonant compared to dissonant chords—according to the Western definition of music: Italian and Himalayan participants were asked to express pleasantness judgments for consonant and dissonant chords. An Italian and a Nepalese sample were tested both at 1,450 m and at 4,750 m of altitude, with the further aim to evaluate the effect of hypoxia on this task. A third sample consisted of two subgroups of Sherpas: lowlanders (1,450 m of altitude), often exposed to Western music, and highlanders (3,427 m of altitude), less exposed to Western music. All Sherpas were tested where they lived.

**Results:** Independently from the altitude, results confirmed the consonance effect in the Italian sample, and the absence of such effect in the Nepalese sample. Lowlander Sherpas revealed the consonance effect, but highlander Sherpas did not show this effect.

**Conclusions:** Results of this pilot study show that neither hypoxia (altitude), nor demographic features (age, schooling, or playing music), nor ethnicity per se influence the consonance effect. We conclude that music preferences are attributable to music exposure.

Corresponding author
Vittore Verratti,
vittore.verratti@unich.it

## INTRODUCTION

The topic of music pleasantness extends towards a plethora of scientific fields, from psychology to neurology, from musicology to sociology, from ethnology to biology (*Cross, 2003*; *Bowling & Purves, 2015*; *Bowling et al., 2017*). The first issue in this domain is the

definition of music pleasantness itself, followed by the difficulty in understanding whether universal laws exist according to which music can be categorized as pleasant vs unpleasant, or it is a personal preference, possibly influenced by culture (*Cazden, 1980*; *Higgins, 2012*; *Harrison & Pearce, 2020*). Concerning the first issue, in Western music pleasantness seems to be linked to vertical harmonicity (*Terhardt, 1984*; *McDermott, Lehr & Oxenham, 2010*; *McDermott et al., 2016*). Harmonicity, in fact, can be referred to either a "vertical" dimension, with isolated chord presentation, or to a "horizontal" dimension, with sequential chords presentation. Here the term harmonicity will be used to refer to vertical harmonicity. Harmonicity is obtained when the pitches in a chord are related by simple integer ratios of frequencies, and in Western culture it would contribute to the perceived consonance and to the following pleasantness of the chord. We can summarize this concept in the following way: a harmonic chord (a set of notes played together) sounds as pleasant if it is consonant (*Bowling & Purves, 2015*). In Western culture, in fact, consonance is associated with pleasantness, stability and relaxation, and dissonance is associated with unpleasantness (the so-called "consonance effect"; *Prete et al., 2015*, *2019*).

A chord is obtained from the overlapping of more pure tones defined as partials. When perceived together, neighboring partials could lead to interference, producing dissonance. Interference is obtained when the spectral components in a chord are close enough to elicit masking and beating, which have been proposed to be the acoustic base for perceived roughness (for a deeper dissertation on different theories about consonance see *Tramo et al., 2001*; *Harrison & Pearce, 2020*). These principles of music consonance seem to be culturally defined, in fact they are valid in Western music, which is based on harmonicity. Differently from Western listeners, who perceive interference as unpleasant, in some non-Western cultures beating is associated to consonance (*Messner, 1981*), showing that the aesthetic appreciation of interference is culturally defined. According to this view, music harmonicity is a central feature of Western music, but it is less diffuse or absent in other cultures (*Malm, 1996*). This leads to the second issue raised above: if harmonicity is the base for the consonance effect in Western music, how are consonant and dissonant chords evaluated in those cultures in which harmonicity is not present? Answering this question can help in defining music preferences as established by nature or by nurture (*Bowling et al., 2017*): on the one hand, some agreement should be expected in the evaluation of pleasant vs unpleasant music across different cultures if music preferences are innate and biologically determined, but–on the other hand–different preferences for different music should be found among cultures if music preference is culturally defined.

Both of the aforementioned hypotheses found support in previous studies. Some reports suggested music consonance to be a universal principle, independent from the cultural environment: for instance, a preference for consonance was described even in different species, such as birds and monkeys (*Fishman et al., 2001*; *Chiandetti & Vallortigara, 2011*), and it has been shown that 2-month-old infants prefer to listen to consonant over dissonant chords (*Trainor, Tsang & Cheung, 2002*). Nevertheless, this evidence was not confirmed in a different study with 6-month-old infants (*Plantinga & Trehub, 2014*), and

in other studies with monkeys (*McDermott & Hauser, 2004*), suggesting that music consonance might not be universal, developing instead with experience and listening habits, namely on culture (*McDermott et al., 2016*). Along similar lines, it has been suggested that the consonance/dissonance spectrum would be context-sensitive to the original ecological listening experience (*Popescu et al., 2019*).

A possible way to empirically disentangle the innate vs cultural origin of music preferences is represented by cross-cultural studies: the comparison among music preferences expressed by individuals belonging to different cultures can help in shedding light on this topic. In this respect, McDermott and colleagues (*McDermott et al., 2016*) provided evidence for a cultural basis of music preferences, showing that music pleasantness is attributable to exposure to musical harmonicity. In particular, they tested Western (US Americans) participants, native Amazonian participants without exposure to Western culture (Tsimane'), and Bolivians living either in the capital city or in a rural town. The results showed that the Tsimane' did not evaluate differently consonant and dissonant chords, the participants from US confirmed the expected consonance effect, whereas the Bolivian groups were placed in the middle, showing a "consonance effect", but less strong than that of US participants. These findings are in accordance with previous studies in which different populations have been tested. For instance, *Maher (1976)* found different "restful/restless" ratings in Canadians and Indians for harmonic intervals, even if *Butler & Daston (1968)* found no difference in dyadic chords preference between Americans and Japanese, leading the authors to conclude that music preference could be dissociated with respect to the perception of consonance. All of these results confirm the usefulness of cross-cultural studies in the investigation of the possible biological substrate of music preferences.

With the present multicultural and globalized world, it is increasingly difficult to explore genuine cross-cultural variations (*Segall et al., 1990*). English language-based elements are indeed a key factor of modern cultural evenness, and this trend did not spare music listening habits. In this regard, the possibilities offered by extreme environmental research projects, such as the present "Kanchenjunga Exploration and Physiology" project, may be really useful. Indeed, the history of altitude physiology still continues to be enriched, especially by the recent efforts made to understand the peculiarities of high-altitude populations (*West, 2016*). For these reasons, in the present study we tested European and Himalayan participants, exploiting the different music cultures and habits of the two samples. Nepal had its declaration of democracy only in 1951, thus only in the last decades it started to open to different cultures, mainly near India and China (*Dinnerstein & Alter, 2018*). In the same year Radio Nepal was created and it broadcasted in Kathmandu alone, being the only radio station until the 1990s. Nepal has its own music culture, even if it is quite heterogeneous (*Henderson, 2002*), also due to the fact that these territories are inhabited by a mix of different populations, sometimes separated by mountains and thus very different from one another, some other times living together (*Greene, 2003*). In particular, in the North-East of the country, Tibetan groups are frequent, including Sherpas, an independent ethnical group coming from Tibet. Nepalese music is a plethora of numerous traditions and presently it is changing deeply in terms of

content and context. Newar music (representing the societies of the Kathmandu valley) is characterized by complex rhythmic cycles (*Prajapati, 2018*). The traditional Nepalese folk music shares with North India and neighbor countries the expression of tonality through melody, from an expanded set of heptatonic scales. Thus, in respect to Western music, it lacks the harmonic function of chords. In the music of North India and neighbor countries, tonal hierarchy is characterized by addition or deletion of tones, deviating from the underlying scale (*Castellano, Bharucha & Krumhansl, 1984*). Tibetan music (representing the Sherpa society) is mainly based in the vowel modification and contouring of tones, without the Western function of chords and with melodies consisting of several melodic patterns and little melodic ranges (*Tsukamoto, 1983*).

Over 100 million persons live at high altitude, throughout Asia, East Africa, and North, Central and South America (*Moore, 2001*): testing these population requires to detect if changes are due to chronic hypoxia exposure or typical culture. In this regard, the notable increase of altitude traveling in last decades opens novel intriguing perspectives. Evaluating the effect of living altitude or testing altitude may allow to focus specifically on cultural habits.

The "Kanchenjunga Exploration and Physiology" project, a scientific expedition that took place mainly in the Himalayas, offered us the possibility to investigate music preferences in both Nepalese and Sherpa samples, and this in turns allowed us not only to compare music preferences of Western (European) and Himalayan participants, but also to further investigate differences between ethnical groups living close to one another (Nepalese and Sherpa), but having different music habits. It is well known that altitude hypoxia negatively affects some aspects of physiology and, specifically, sensory systems (*Cingi, Erkan & Rettinger, 2010*; *Jha, 2012*; *Ruffini et al., 2015*); regarding pleasantness, little is known about the role of hypoxia: for example, no effect was found for thermal comfort (*Golja et al., 2005*), no effect for flavors pleasantness, and specific effects were found for the evaluation of pleasantness of individual odors due to hypoxia (*Huppertz et al., 2018*). To our knowledge, there is still a lack of results regarding the effect of altitude hypoxia on preferences related to music pleasantness. As well, cognitive and emotional adaptations were reported during altitude volitional exposure or during hypoxia confinement, suggesting that high motivation may drive pleasantness of the experience, even with highly demanding hypoxic exposure (*Karinen & Tuomisto, 2017*; *Stavrou et al., 2018*). In addition, despite the fact that the effect of music on mood has been extensively studied, also in their modulation of stress response (*Koelsch et al., 2016*), few studies, if any, investigated the modulation of mood in specific measures of music pleasantness. We therefore saw, among the other research lines of the project, the special possibility to explore the above-mentioned topics.

## Aim

This study aimed to evaluate cultural variation in music consonance/dissonance-derived pleasantness, a possible role of altitude and a possible mediation by mood, age, being a musician and Western music listening habit. To this aim, we exploited a musical test already used to describe the consonance effect in Western participants (*Prete et al., 2015*),

with three samples belonging to different cultures: Italians, Nepalese and Sherpas. Italian and Nepalese samples were tested both at lower and higher altitudes with the aim to evaluate the possible effect of hypoxia at high altitude. Moreover, a mood questionnaire was also administered to the Italian sample, in order to investigate the effect of hypoxia also on this variable. Starting from the results described by *McDermott et al. (2016)*, we expected to confirm the consonance effect in the Italian group, and likely in lowlander Sherpas, often exposed to Western music, and we did not expect to find such effect in the other groups. Concerning altitude, due to the absence of specific evidence in this regard, we did not have specific expectations on the effect of hypoxia on music pleasantness evaluation; however, considering previous evidence about the role of hypoxia on sensorial functions and pleasantness, we may hypothesize hypoxia may affect somehow music pleasantness. Moreover, we also took into account the possible effects of time spent listening to music, frequency of exposure to Western music, years of education and playing music or otherwise on the consonance effect.

## MATERIALS AND METHODS

### Participants

The task was carried out by three groups of participants ($N = 22$; see Table 1 for demographic information self-reported by each participant): Italians, Nepalese and Sherpas. The Italian group was composed of six healthy Caucasian lowlanders (five males, one female; age: 43.83 ± 15.30 years; BMI: 25.81 ± 3.25 Kg/m$^2$; schooling: 18.50 ± 3.27 years), who were used to listen to Western music, on average 1.5 h/day. The Italian group spoke on average 1.33 foreign languages (range: 0–2, all Western languages). The Nepalese group was composed of six healthy porters (workers who carry loads during altitude expeditions), lowland dwellers (all males; age: 30.33 ± 8.55 years; BMI: 24.36 ± 4.70 Kg/m$^2$; schooling: 5.86 ± 4.71 years), who declared to listen on average 1.37 h/day, but only one participant reported to listen to Western music sometimes. This group also reported to speak on average 1.33 foreign languages (range: 0–3, including three participants who spoke English; the other spoke Eastern languages). The third group was composed by ten healthy Sherpa (all males). This group was further divided into five lowlanders (age: 28.60 ± 5.23 years, schooling: 11.20 ± 3.63 years) and five highlanders (age: 37.00 ± 16.51 years, schooling: 6.40 ± 4.28 years). Lowlander Sherpas were used to spend much time to listen music (on average, 4 h/day) and, except one participant, they declared to be often exposed to Western music. This seems to be in line with the self-reported data according to which all of them spoke at least three foreign languages, included English (range: 3–4, average: 3.2). Highlander Sherpas listened to less music (0.9 h/day) and only one participant in this subsample stated to sometimes listen to Western music. Only this participant and another one in this subsample declared to speak English, with a mean of 1.6 foreign languages spoken in the subsample (range: 0–3; Eastern languages). Among all participants, seven of them played music since at least 3 years (two Italian, one Nepalese, two lowlander and two highlander Sherpas). We decided to consider three main groups according to the participants' ethnicity (Italian, Nepalese and Sherpa), independently of their reported exposure to Western music, knowledge of Western

**Table 1 Self-reported demographic data.**

| N | Eth | Liv | Test | Sex | Age | Sch | Mus | Y/Mus | West | Freq | Lang | Eng | Hand |
|---|-----|-----|------|-----|-----|-----|-----|-------|------|------|------|-----|------|
| 1 | ITA | L | L-H | F | 36 | 21 | YES | 30 | YES | often | 2 | YES | R |
| 2 | ITA | | | M | 63 | 18 | NO | 0 | YES | always | 0 | NO | R |
| 3 | ITA | | | M | 59 | 13 | NO | 0 | YES | always | 1 | NO | R |
| 4 | ITA | | | M | 25 | 17 | NO | 0 | YES | always | 1 | YES | R |
| 5 | ITA | | | M | 32 | 20 | YES | 10 | YES | often | 2 | YES | R |
| 6 | ITA | | | M | 48 | 22 | NO | 0 | YES | always | 2 | YES | R |
| 7 | NEP | L | L-H | M | 26 | 12 | NO | 0 | YES | little | 3 | YES | R |
| 8 | NEP | | | M | 18 | 10 | NO | 0 | NO | never | 0 | NO | R |
| 9 | NEP | | | M | 39 | 0 | NO | 0 | NO | never | 0 | NO | R |
| 10 | NEP | | | M | 40 | 7 | NO | 0 | NO | never | 2 | YES | R |
| 11 | NEP | | | M | 30 | 4 | YES | 3 | NO | never | 2 | YES | L |
| 12 | NEP | | | M | 29 | 0 | NO | 0 | NO | never | 1 | NO | R |
| 13 | SHEh | H | H | M | 56 | 7 | NO | 0 | NO | never | 3 | YES | R |
| 14 | SHEh | | | M | 23 | 12 | YES | 7 | YES | often | 3 | YES | L |
| 15 | SHEh | | | M | 25 | 6 | NO | 0 | NO | never | 0 | NO | R |
| 16 | SHEh | | | M | 54 | 0 | NO | 0 | NO | never | 1 | NO | R |
| 17 | SHEh | | | M | 27 | 7 | YES | 6 | NO | never | 1 | NO | R |
| 18 | SHEl | L | L | M | 37 | 16 | YES | 5-10 | YES | often | 3 | YES | R |
| 19 | SHEl | | | M | 23 | 12 | YES | 5-10 | YES | often | 3 | YES | R |
| 20 | SHEl | | | M | 29 | 6 | NO | 0 | YES | often | 4 | YES | R |
| 21 | SHEl | | | M | 28 | 12 | NO | 0 | NO | never | 3 | YES | R |
| 22 | SHEl | | | M | 26 | 10 | NO | 0 | YES | often | 3 | YES | R |

Note:
Self-reported demographic data: Eth (Ethnicity: ITA = Italian, NEP = Nepalese, SHEh = highlander Sherpas; SHEl = lowlander Sherpa); Liv (Living altitude: L = Low altitude, H = High altitude); Test (Testing altitude: L = Low altitude, H = High altitude); Sex (F = female, M = male); Age (years); Sch (schooling: years of instructions); Mus (YES = musician, NO = not musician); Y/Mus (years of music study); West (YES = listening to Western music; NO = not listening to Western music); Freq (frequency of Western music listening); Lang (number of foreign language spoken); Eng (Knowledge of English language: YES = English speaker, NO = non-English speaker); Hand (handedness: L = left, R = right).

languages, and the other demographic features, in order to investigate the pure effect of ethnicity on the music pleasantness evaluation. Table 1 summarizes demographic data of the whole sample. No participant reported symptoms of Acute Mountain Sickness during the trek.

## Design of the study

The research project "Kanchenjunga Exploration and Physiology" was a subset of "Environmentally-modulated metabolic adaptation to hypoxia in altitude natives and sea-level dwellers: from integrative to molecular (proteomics, epigenetics and ROS) level" approved by the Ethical Review Board of the Nepal Health Research Council (NHRC, reference number 458). All study procedures were performed in accordance with the ethical standards of the 1964 Helsinki declaration and its later amendments or comparable ethical standards. All participants provided their written informed consents.

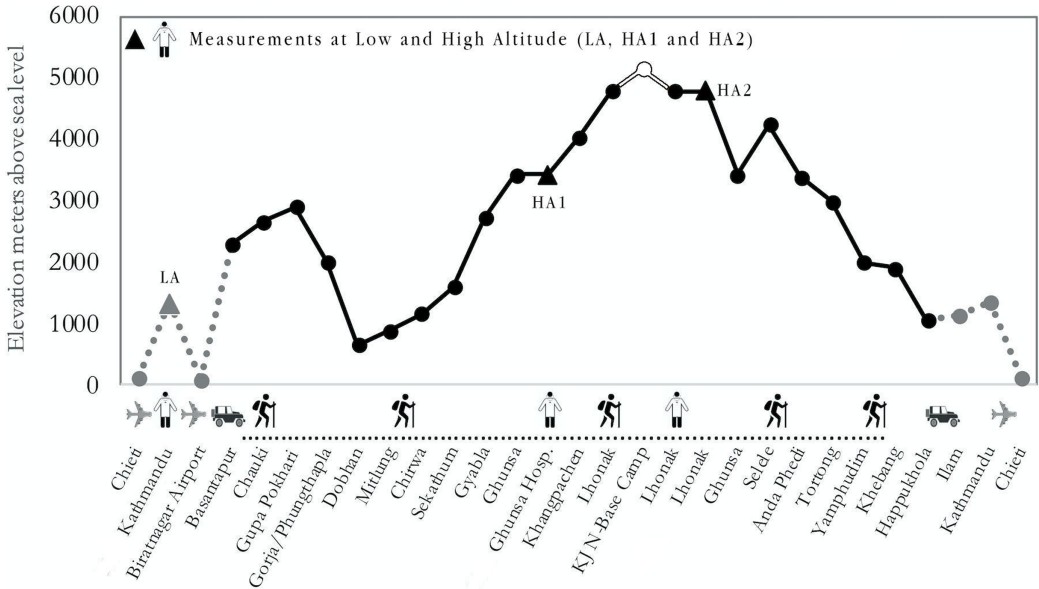

**Figure 1 Altimetric plan of "Kanchenjunga Exploration and Physiology" project.** The continuous line represents the effective trek. The plain data points represent the nights spent in the related location. The empty circular data point at Kanchenjunga North Base Camp (KJ N-Base Camp) represents the highest altitude reached by participants throughout the trek. The triangular data points represent the measurements.

The expedition consisted of a combined circuit of 300 Km distance (south and north base camps), covering a daily average of 6 hours walk, for a total of 110 h, along a demanding route with ascent and descent covering totally over 16,000 m in altitude, in the Himalayan mountain range of eastern Nepal, at the border with Sikkim (India). The project investigated adaptive physiological responses during a trekking at moderate and high altitude, in different experimental groups (Italians and Nepalese, for more details see "Participant" section).

The study protocol involved music pleasantness testing on Italian (group 1) and Nepalese (group 2) participants at lower and higher altitudes (respectively LA and HA2, see Fig. 1). Italians and Nepalese carried out the same auditory test twice, in Kathmandu (1,450 m) and in Lhonak (4,780 m). Only Italian participants also underwent mood state testing on the same days, at the end of the auditory test. Sherpas were tested once, in a between-subjects design: lowlander Sherpas were tested in Kathmandu, where they lived, whereas highlander Sherpas were tested in Ghunsa (HA1, see Fig. 1) at 3,427 m, where they lived. A native Nepalese speaker assisted during the test, both in Kathmandu, Ghunsa and Lhonak, confirming participants understood the task.

## Stimuli and procedure

The stimulus set was composed of 24 triad chords, including 12 consonant and 12 dissonant chords, according to the Western music definition of consonance and dissonance. The same set of stimuli and paradigm had already been used with an Italian sample (*Prete et al., 2015*), confirming the "consonance effect", namely the lower and
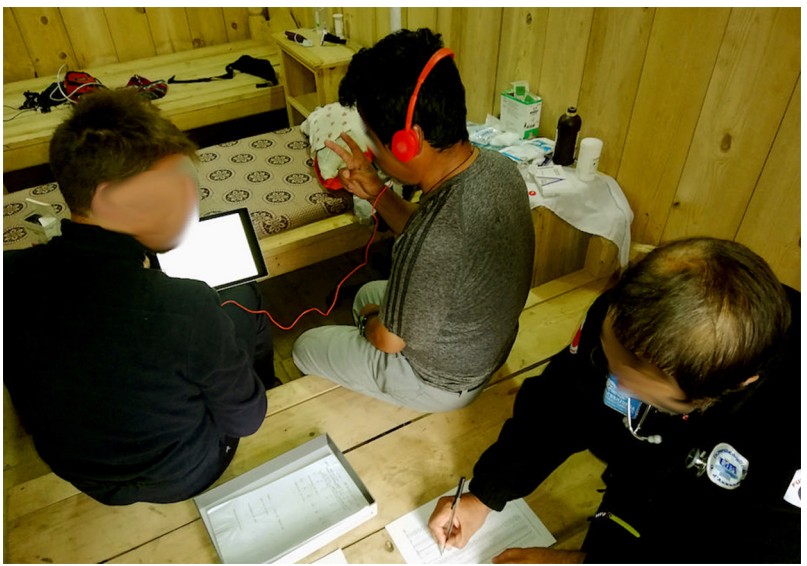

**Figure 2 Actual experimental setting at high altitude.**

higher pleasantness evaluation of dissonant and consonant chords, respectively. Consonant chords consisted of major third intervals (interval ratio 4:5), perfect fiveth intervals (interval ratio 2:3) and minor third and sixth intervals (interval ratio 5:6 and 5:8, respectively); dissonant chords consisted of minor and major second intervals (interval ratio 15:16 and 8:9) and in minor and major seventh intervals (interval ratio 9:16 and 8:15). By exploiting these intervals, three consonant and three dissonant triads were created with a piano timbre and, by means of GoldWave v5.25 software (GoldWave Inc., St. John's, NL, Canada): in particular, in the consonant chords all intervals among three notes were consonant (C – E♭ – G; C – E – G; C – E - A♭), and in the dissonant chords all intervals among three notes were dissonant (B♯ - C♯ - D; C – D♭ – B; C - A♯ - B). Then, these six chords were transposed up by 4, 6 and 9 semitones, obtaining four chords for each of the original ones: the position of the tonic of each chord was transposed, and the distance among notes remained the same. In this way, 12 consonant and 12 dissonant chords were created (stimuli are available as Supplemental Materials). Each stimulus lasted 1,330 ms and included 630 ms of linear fade out and was repeated four times for a total set of 96 stimuli, presented in a random order by means of headphones (see Fig. 2).

The paradigm was presented by means of a tablet PC (Microsoft Surface Pro 6) and it was controlled by the experimenter, who sat behind the participant for the whole task. Before the beginning of the task, experimental instructions were verbally presented: participants were instructed that very brief audio clips would be presented one at time, and that they will be required to express a pleasantness judgment for each clip, by using a four point-Likert scale (1 = not at all; 2 = little; 3 = enough; 4 = very much). The list of four responses were also printed in the mother tongue of each participant in order to avoid possible confounding effects on the scale used, and to make sure that the response provided by the participant could be clearly recognized by the experimenter. Once the
response was provided, the experimenter recorded the number corresponding to the response on a sheet previously prepared, and started the next trial.

After the end of the auditory task, only Italian participants were required to fill the 58-item Italian version of Profile Of Mood State (POMS). In this questionnaire, a global score, called Total Mood Disturbance (TMD), is calculated from the sub-scores as follows: Tension + Depression + Anger + Fatigue + Confusion − Vigour + 100. This type of testing had already been used in monitoring the mood state in sports and hypoxic environments (*Peri, Scarlata & Barbarito, 2000*; *Kenttä, Hassmén & Raglin, 2006*; *Karinen & Tuomisto, 2017*).

## Statistical analyses

Statistical analyses were carried out using the R-based open-source software Jamovi (The Jamovi project. jamovi (Version 1.0) (Computer Software). https://www.jamovi.org 2019) for General Linear Model (GLM) Mediation Analyses, and by means of Statistica 8.0.550 (StatSoft. Inc., Tulsa, OK, USA) for Analyses of Variance (ANOVAs).

Mean pleasantness evaluations for consonant and dissonant chords expressed at the lower altitude (Kathmandu) and at the higher altitude (Ghunsa and Lhonak) were considered separately. Moreover, the difference of pleasantness evaluation between consonant and dissonant chords was also calculated, obtaining an overall measure in which negative scores corresponded to dissonance preference, positive scores corresponded to consonance preference and a score of 0 corresponded to an equal preference between consonant and dissonant chords.

A first GLM Mediation Analysis was used to test the mediation role of age and years of schooling, with Consonance–Dissonance difference, Consonance pleasantness and Dissonance pleasantness as the dependent variables and Ethnicity (Italian, Nepalese, Sherpa), Listening to Western music (yes vs no) or Playing music (yes vs no) as the independent variables. A second GLM Mediation Analysis was used on the Italian sample to test TMD (from POMS questionnaire) as a mediator, with Consonance pleasantness, Dissonance pleasantness, and Consonance-Dissonance difference as the dependent variables and Altitude (low, high) as the independent variable. Both mediations were carried out with the jAMM (Advanced Mediation Models) suite for Jamovi software (https://jamovi-amm.github.io/index.html).

Then, in a mixed-model ANOVA, Ethnicity (Italian, Nepalese, Lowlander-Sherpa) was considered as a between-subjects factor and Chord (consonant, dissonant) was used as a within-subject factor. Pleasantness evaluations recorded at the lower altitude were used as the dependent variable. In a second ANOVA, Ethnicity (Italian, Nepalese, Highlander-Sherpa) was considered as a between-subjects factor, Chord (consonant, dissonant) was used as a within-subject factor, and pleasantness evaluations recorded at the higher altitude were used as the dependent variable. This separation was necessary because Italian and Nepalese participants carried out the task at both altitudes, whereas Sherpa participants who carried out the task at lower and higher altitudes were two different subgroups. For this reason, in a third ANOVA, only Italian and Nepalese participants were included, and both Chord (consonant, dissonant) and Altitude

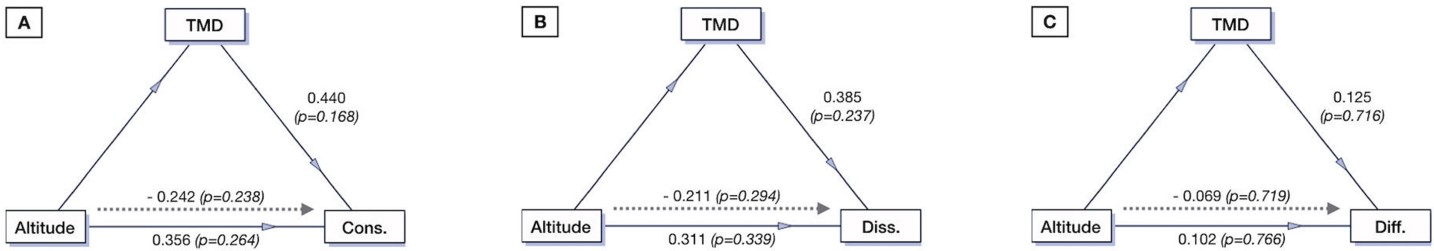

**Figure 3 Mediation analysis of Total Mood Disturbance on the relation between altitude and music pleasantness.** Mediation analysis of Total Mood Disturbance (TMD) from Profile Of Mood State (POMS) questionnaire, on the relation between altitude and music pleasantness. Values displayed are β weights. Dotted lines represent the indirect effects (i.e. those mediated by TMD), while the whole lines represent the direct effects. (A) Shows the analysis on Consonance (Cons.), (B) on Dissonance (Diss.) and (C) on the difference Consonance–Dissonance (Diff.).

(low, high) were used as within-subject factors, whereas Ethnicity (Italian, Nepalese) was used as between-subject factor. Pleasantness evaluations were used as the dependent variable. Finally, considering only the Sherpa subgroups, a fourth ANOVA was carried out, by using the pleasantness evaluation as the dependent variable, and by using Chord (consonant, dissonant) as within-subject factor and Altitude (low, high) as between-subject factor. In all ANOVAs post-hoc tests were carried out by means of Duncan test.

## RESULTS

The first GLM Mediation Analysis revealed that, among the variables of Ethnicity, Playing music and Listening to Western music, with Age and Schooling as mediators, the only one that significantly predicted Consonance-Dissonance difference was Listening to Western music (β weight = 0.572, $p < 0.001$), mainly and significantly with a direct effect (β = 0.441, $p = 0.021$), not with the mediation of Age (β = 0.010, $p = 0.813$) nor Schooling (β = 0.121, $p = 0.241$). Listening to Western music also predicted Consonance pleasantness (β = 0.706, $p = 0.009$) without the mediation of Age or schooling, whereas did not predict Dissonance pleasantness (β = 0.463, $p = 0.219$).

The second GLM Mediation Analysis, carried out to test the role of TMD (from POMS questionnaire), revealed that altitude significantly predicted TMD (β = −0.550, $p = 0.023$), with lower TMD values at high (93.50 ± 9.16) with respect to low altitude (115.83 ± 24.65). No significant prediction was found for TMD with respect to any of the predictor variables (Consonance, Dissonance, Difference between consonance and dissonance), nor was any mediation revealed (see Fig. 3); as aforementioned, the TMD test was only done for the Italian group.

In the first ANOVA (lower altitude), the main effect of Chord was significant ($F_{(1, 14)} = 40.62$, $p < 0.001$, $\eta_p^2 = 0.74$), confirming the "consonance effect", with higher pleasantness scores for consonant chords ($M \pm SEM: 2.63 \pm 0.15$) with respect to dissonant chords (2.1 ± 0.15). Importantly, the interaction between Ethnicity and Chord was significant ($F_{(2, 14)} = 12.58$, $p < 0.001$, $\eta_p^2 = 0.64$) and post-hoc tests confirmed the consonance effect in both Italian and Sherpa samples ($p < 0.001$ for both comparisons), but not in Nepalese participants ($p = 0.82$).

In the second ANOVA (higher altitude), the main effect of Chord did not reach significance ($F_{(1, 14)}$ = 3.53, $p$ = 0.081). The main effect of Ethnicity was significant ($F_{(2, 14)}$ = 3.82, $p$ = 0.048, $\eta_p^2$ = 0.35), revealing lower pleasantness scores for Sherpa (1.85 ± 0.25) with respect to both Italian (2.45 ± 0.25, $p$ = 0.028) and Nepalese (2.46 ± 0.08, $p$ = 0.033) participants. Finally, the interaction was also significant ($F_{(2, 14)}$ = 10.88, $p$ = 0.001, $\eta_p^2$ = 0.61), and post-hoc comparison showed that only in the Italian sample consonant chords (2.94 ± 0.31) were judged as more pleasant than dissonant chords (1.97 ± 0.29, $p$ < 0.001). Furthermore, consonant chords received lower pleasantness judgments by Sherpa (1.74 ± 0.17) compared to both Italian (2.94 ± 0.31, $p$ < 0.001) and Nepalese (2.42 ± 0.12, $p$ = 0.038) participants.

The third ANOVA, carried out considering lower and higher altitude scores of the Italian and Nepalese groups, further confirmed the consonance effect, as shown by the main effect of Chord ($F_{(1, 20)}$ = 24.64, $p$ < 0.001, $\eta_p^2$ = 0.55), with higher pleasantness scores for consonant (2.61 ± 0.12) than for dissonant (2.17 ± 0.11) chords. Importantly, the interaction between Ethnicity and Chord was significant ($F_{(1, 20)}$ = 32.33, $p$ < 0.001, $\eta_p^2$ = 0.62; Fig. 4), and post-hoc comparisons showed that the consonant effect was due exclusively to the Italian group ($p$ < 0.001), who rated consonant chords as more pleasant than dissonant chords. Accordingly, consonant chords were evaluated as more pleasant by Italian than Nepalese participants ($p$ = 0.03), whereas dissonant chords were evaluated as less pleasant by Italian than Nepalese participants ($p$ = 0.025). The other main effects and interactions were not significant, revealing no effect of the altitude on the auditory test.

The fourth ANOVA, carried out on the pleasantness scores of the two Sherpa groups tested either at lower or higher altitude, showed a trend toward the consonant effect which failed to reach statistical significance ($F_{(1, 8)}$ = 3.23, $p$ = 0.056). Only the interaction between Altitude and Chord was significant ($F_{(1, 20)}$ = 17.22, $p$ = 0.003, $\eta_p^2$ = 0.68; Fig. 5), revealing a consonance effect only in the lowlander group ($p$ = 0.002), but not in the highlander group ($p$ = 0.21). Furthermore, consonant chords were judged as more pleasant by lowlanders than by highlanders ($p$ = 0.018).

## DISCUSSION

The first important finding of the present cross-cultural pilot study supports the cultural origin of music pleasantness: the consonance effect was confirmed in the Italian sample and it was not found in the Nepalese sample. This first result would exclude the possibility that preference for consonant music is innate and biologically determined, because if this hypothesis were true we should have found a similar preference in all the samples tested. We were able to test this hypothesis because in Nepalese and Sherpa groups there were participants who declared not to listen to Western music. While it can be assumed for Highlander Sherpas, living in remote regions, one may be wondering why there were diverse habits between lowlander Sherpas and Nepalese. We think the reason of the difference to be related to the working habit: if Sherpas living in Kathmandu were used to be involved directly in commercial or mountaineering activities with foreign

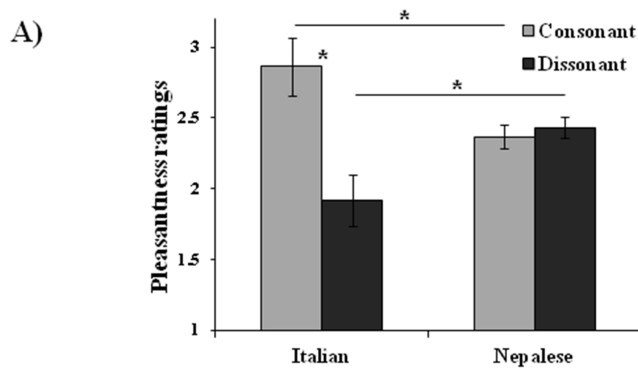

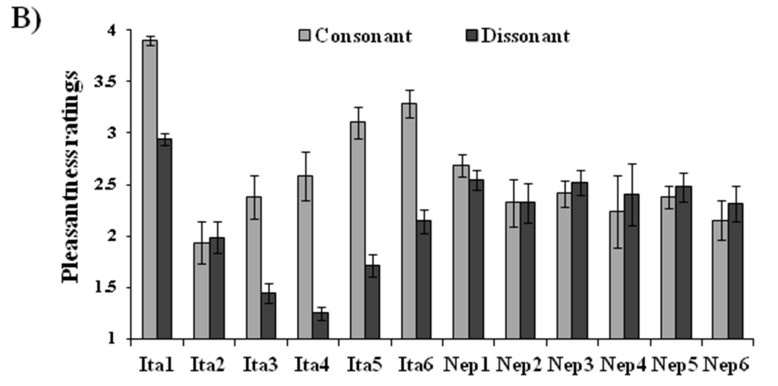

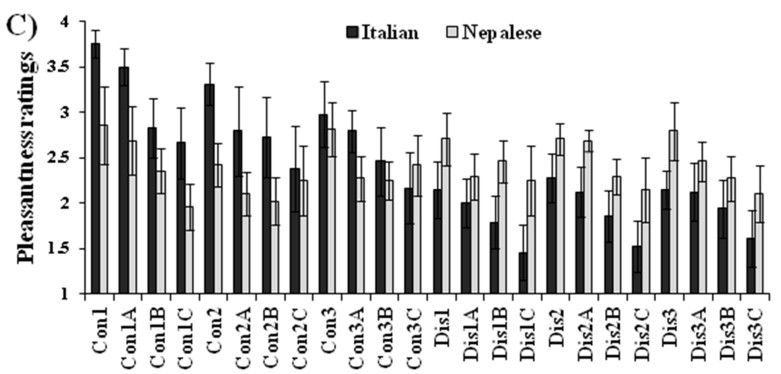

**Figure 4 Interaction between Ethnicity and Chord.** (A) Interaction between Ethnicity (Italian, Nepalese) and Chord (consonant, dissonant), on the pleasantness judgments; asterisks show significant comparisons ($p < 0.05$); (B) pleasantness judgments for consonant and dissonant chords expressed by each participant of the two groups; (C) pleasantness judgments by Italian and Nepalese participants expressed for each chord. Judgments were expressed on a 1–4 Likert scale. Bars represent standard errors.

individuals, Nepalese porters had a secondary role in the link with mountaineers or trekkers.

Importantly, a further crucial result was obtained in the group of Sherpas: the results showed that a consonance effect was present in lowlander Sherpas, namely the subsample who declared to listen to Western music, but the effect was completely absent in highlander Sherpas, the subsample who declared to listen to music less frequently and,

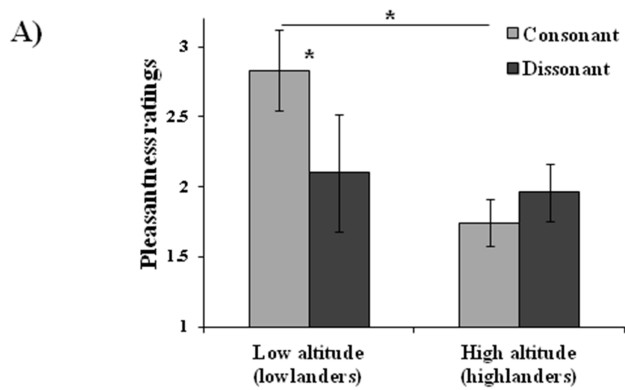

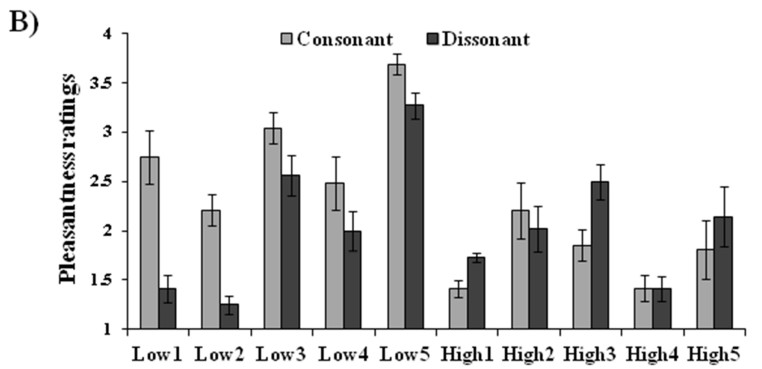

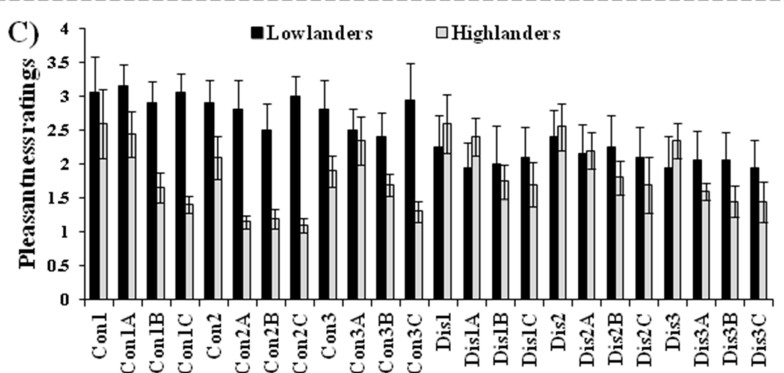

**Figure 5 Interaction between Altitude and Chord.** (A) Interaction between Altitude (low altitude: lowlander subsample, high altitude: highlander subsample) and Chord (consonant, dissonant), on the pleasantness judgments; asterisks show significant comparisons ($p < 0.05$). (B) Pleasantness judgments for consonant and dissonant chords expressed by each participant of the two subgroups; (C) pleasantness judgments by lowlander and highlander Sherpas expressed for each chord. Judgments were expressed on a 1–4 Likert scale. Bars represent standard errors.

importantly, not to listen to Western music. This latter result is the most important of the present study, because it shows that music exposure is the key factor determining the consonance effect: even if they belong to the same ethnicity, only Sherpas often exposed to Western music showed a consonance effect, even if to a lesser degree than the Italian sample. This conclusion is statistically confirmed also by the mediation analysis, revealing that listening to Western music influenced the consonance effect, independently from the

ethnical group. Indeed, listening to Western music predicted consonance–dissonance difference, as well as consonance pleasantness, and did not predict dissonance pleasantness. In addition, neither age nor schooling mediated these results, allowing us to affirm that the only effect which influenced the consonance effect was the exposure to Western music.

In the present study we also explored the possible effect of some demographic features on the consonance effect, such as age, schooling and playing music: the results revealed that none of these variables influences music pleasantness, stressing that the exposure to a particular kind of music is likely the most salient variable responsible for the consonance effect. It has to be underlined that previous evidence revealed that preference for consonance is related to musical experience, showing a higher preference for harmonic chords in participants who had music training, even if aesthetic judgments for dissonant chords were not influenced by musical training (*McDermott, Lehr & Oxenham, 2010*). Moreover, in a cross-cultural study with Western and African (Cameroon) participants, Fritz and colleagues (*Fritz et al., 2009*) found no group differences in judging both emotional content (happy, sad, fearful) and pleasantness of Western and African music. Similarly, no difference in chords preference between American and Japanese listeners were found by *Butler & Daston (1968)*, whereas a preference for consonance vs dissonance was described in newborns (*Trainor, Tsang & Cheung, 2002*) as well as in non-human species (*Fishman et al., 2001*; *Chiandetti & Vallortigara, 2011*), thus suggesting a biological basis of music processing, independent from the culture.

Even if we did not have specific hypotheses in this regard, we also considered altitude hypoxia as a possible influencing variable, because of the well-known affecting role of such condition in sensory systems and cognitive performance. Indeed, among the numerous evidence on the effects of hypoxia, nasal, ear, and throat complaints (*Cingi, Erkan & Rettinger, 2010*), short-term and long-term eye diseases (*Jha, 2012*), increases of the olfactory threshold (*Ruffini et al., 2015*), and impairment of attentional performances (*Limmer & Platen, 2018*) have been shown. Instead, our results showed that music pleasantness was not influenced by altitude.

Other mood dimensions were affected by hypoxia, as resembled by the POMS: the Italian sample revealed lower mood disturbance scores (index of tension, depression, anger, fatigue and confusion) at higher compared to lower altitudes. We speculate that this difference could be due to the fact that once climbed the mountain, the more positive mood can be due to a kind of gratification compared to the possible concern before the climb. This result agrees with the findings of *Karinen & Tuomisto (2017)*, who reported that during a prolonged expedition in Everest mountain well-motivated participants were capable to maintain a good mood state; on the contrary, *Stavrou et al. (2018)*, who reported that hypoxia was capable to negatively affect the mood, exacerbating the effect of bed rest and ambulatory confinement. Other evidence linked mood and altitude hypoxia, affirming that mood states were adversely affected by duration and level of altitude, but with a main role of physical exertion (*Shukitt-Hale & Lieberman, 1996*). Physical exercise is known to affect mood state (*Di Corrado et al., 2014*). In this regard, *Murgia et al. (2016)* recently found that physical exertion alone may negatively affect mood

states, mainly in low-performing athletes. Compared to these latest results, participants of the present study were not particularly affected by physical exertion, nor they experienced an unsuccessful performance. Several physiological and psychological variables can support the lower mood disturbance found at high altitude in the current study. According to the self-efficacy hypothesis, gratification might have played a role: the degree of reward, combined with an athletic success such as trekking to a Himalayan base camp, would positively have affected the mood state (*Eldar et al., 2018*). Among the biochemical factors, endorphins might be evoked—considering the positive association with long term high altitude exposure (*Appenzeller & Wood, 1992*)—as well as modulations of hypothalamic-pituitary-adrenal axis, mTOR signaling and serotonin release (*Mikkelsen et al., 2017*). Moreover, Heinrich and colleagues recently affirmed that acclimatization plays a big role in cognitive and mood alterations during a sojourn at altitude (*Heinrich et al., 2019*). To be noted that none of our participants suffered remarkable altitude sickness symptoms. In summary, neither strenuous physical exertion, nor ineffective acclimatization, nor forced confinement stressed our participants. Thus, from these findings and from our results, we support the idea that hypoxia may increase the mood disturbance during non-rewarding conditions or in association with altitude sickness symptoms or strenuous physical exercise, but that hypoxia per se, or hypoxia during motivating conditions does not promote mood disturbances.

The present results, suggesting a crucial role of music exposure on music pleasantness, are largely in accordance with those described by *McDermott et al. (2016)* who tested US, Bolivian and Amazonian participants. In their study, the authors found that Western participants judged consonant chords as more pleasant than dissonant chords, and they found that Amazonians—not exposed to Western culture—did not show this preference, and Bolivians' scores were halfway. This pattern of results is confirmed also by the present study, in which the preference for consonant over dissonant chords was high for Italians, it was absent for Nepalese and Sherpa not exposed to Western culture, but it was present (in a lesser degree compared to Italians) in Sherpa often exposed to Western music. Compared to their findings, the present study adds the evidence of null effects of other demographic information on music preference, as well as it shows a null effect of hypobaric hypoxia on the pleasantness of music.

We should highlight that, due to the low sample size tested in this pilot study, which also led to an incomplete design (for instance, not all of the three groups were tested both at lower and higher altitude), and to the absence of control tasks, caution is needed in the generalization of the present results. Nevertheless, we should also highlight that our findings strictly resemble those recently described by *McDermott et al. (2016)*. It has to be noted, moreover, that the relatively low sample size is also due to the difficulty in testing participants belonging to remote cultures, and mainly to the fact that the test was administered at different altitudes, including Lhonak (4,780 m); the same contextual factors limited the possibility to perform control tasks and to test a full factorial design (i.e., each group of Sherpas was tested where participants lived, and we cannot obtain data of lowlanders/highlanders at high/low altitude, respectively). However, these limitations are typical of field studies in extreme environments. In this frame, we underlie the need of

more cross-cultural evidence in order to assess the possible origin of music pleasantness. Starting from the results collected in this pilot study, we can exclude the effect of hypoxia on pleasantness evaluation, and this result will allow to avoid one of the aspect which made it difficult to enlarge the sample, namely altitude, allowing to focus on music habits (i.e., listening to Western music) more than on living altitude or testing altitude. Similarly, the fact that the mood seems to be related only to altitude (hypoxia) allows us to conclude that music pleasantness is not influenced by this variable, but also in this case caution is needed because, due to a technical issue, POMS was administered only to the Italian sample. Despite we did not carry out a control task, our Nepalese sample demonstrated to have understood the instructions, from a brief familiarization with the experimental setting and feedback from the native speakers who supported us translating the instructions.

## CONCLUSION

To summarize, we could consider these pilot results as a further evidence in favor of the crucial role of music exposure in music preference, having found that neither physiological changes possibly due to altitude, nor demographic differences influence this preference. Cross-cultural studies represent unique opportunities to test the biological bases of the most disparate human behaviors, and the present results add further evidence supporting that music pleasantness is not universally shared, but strongly depends upon habits.

### Funding
This work was supported by a grant from the Department of Psychological, Health and Territorial Sciences, "G. d'Annunzio" University of Chieti-Pescara, Italy, to Vittore Verratti. The funders had no role in study design, data collection and analysis, decision to publish, or preparation of the manuscript.

### Grant Disclosures
The following grant information was disclosed by the authors:
Department of Psychological, Health and Territorial Sciences, "G. d'Annunzio" University of Chieti-Pescara, Italy, to Vittore Verratti.

### Competing Interests
Luca Tommasi is an Academic Editor for PeerJ

### Author Contributions
- Giulia Prete conceived and designed the experiments, analyzed the data, prepared figures and/or tables, authored or reviewed drafts of the paper, and approved the final draft.
- Danilo Bondi conceived and designed the experiments, performed the experiments, analyzed the data, prepared figures and/or tables, authored or reviewed drafts of the paper, and approved the final draft.

- Vittore Verratti conceived and designed the experiments, performed the experiments, authored or reviewed drafts of the paper, and approved the final draft.
- Anna Maria Aloisi conceived and designed the experiments, authored or reviewed drafts of the paper, and approved the final draft.
- Prabin Rai conceived and designed the experiments, performed the experiments, authored or reviewed drafts of the paper, and approved the final draft.
- Luca Tommasi conceived and designed the experiments, authored or reviewed drafts of the paper, and approved the final draft.

## Human Ethics

The following information was supplied relating to ethical approvals (i.e., approving body and any reference numbers):

The Nepal Health Research Council (NHRC) granted Ethical approval to carry out the study (Ethical Application Ref: 458).

The present study did not involve patients, children or animals, as well as drugs, genetic samples or invasive techniques, thus it was not subject to ethical review by the academic medical research board in Italy.

## Data Availability

The raw measurements are available in the Supplemental Files.

## Supplemental Information

Supplemental information for this article can be found online at http://dx.doi.org/10.7717/peerj.9344#supplemental-information.

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
