# Peer review of "Universality vs experience: a cross-cultural pilot study on the consonance effect in music at different altitudes"

_PeerJ, doi:10.7717/peerj.9344_

## Round 0.1 · original submission · Major Revisions

Please look carefully at and address all of the Reviewers' comments, and I would be happy to reconsider your paper.

Reviewer 1 ·

Basic reporting

The language is generally clear, though there are various passage where the paper could benefit from some careful proofreading. Examples:
• Abstract: ‘consisting in’ should be ‘consisting of’
• L 66.: ‘infants prefer listen’ should be ‘infants prefer to listen’
• L 125: ‘(see(Prete et al., 2015))’ should be ‘(see Prete et al., 2015)’
• Throughout the manuscript, the authors use the word ‘exposition’ when really they mean ‘exposure’. These two words have different meanings.

The article is structured appropriately and has appropriate figures and tables. It is generally self-contained with relevant results to the hypotheses.

I’m happy to see that the authors have shared their data. Ideally, though, this data would include stimulus-level responses, rather than just participant-level condition means. It would be a shame if this information is no longer available.

The theoretical background to consonance perception is not explained well. In particular, the description of how “harmony” may contribute to perceptual consonance is very unclear. It would be better to use the term “harmonicity”, because “harmony” has a special meaning in Western music theory that is different to how the authors use the word. It may also be worth mentioning the potential role of interference between partials (or roughness, or beating) in consonance perception. For a theoretical review, see the recent article “Simultaneous consonance in music perception and composition” by Harrison and Pearce.

To the extent that consonance perception is culturally determined, we would expect the consonance judgements of the Nepalese to reflect the distributions of pitch content in their musical style. However, the authors do not provide any account of the pitch structure of Nepalese music, or explain the way in which it deviates from Western music. This kind of detail is essential background for understanding the nature of consonance perception in Nepalese listeners, especially if the authors subscribe to a cultural account of consonance perception.

It is useful that the authors thought of hypoxia as a potential moderator of consonance judgements in this study. It would be sufficient to present this as a control analysis rather than a primary research question. However, given that the current paper considers hypoxia to be a primary research question, there needs to be more theoretical motivation in the introduction for why hypoxia might influence consonance judgements. It is currently not clear to me why one would anticipate that hypoxia would affect the relative consonance of musical chords. This discussion could be grounded in the three theories of consonance (harmonicity, interference, and culture) discussed earlier.

There are a couple of other historic cross-cultural studies of consonance perception that should be mentioned in the introduction:

> Butler, J. W., & Daston, P. G. (1968). Musical consonance as musical preference: A cross-cultural study. The Journal of General Psychology, 79, 129-142.

> Maher, T. F. (1976). “Need for resolution” ratings for harmonic musical intervals: A comparison between Indians and Canadians. Journal of Cross-Cultural Psychology, 7, 259-276.

The bar plots should be adjusted such that the lower tails of the error bars are visible.

The mediation analyses are a nice touch, but in other senses the statistical analyses and plots deserve to be extended. It would be useful to have a plot showing consonance judgements split by stimulus and by participant group, as in McDermott et al. (2016). This would help the reader to better understand the nature of the participants’ consonance judgements. It would be useful to replace the bar plots with figures that include raw data points, such that the reader can better understand the variability in the sample. It would also be useful to modify these was plots so that they reflect the repeated measures structure of the design, for example by plotting the difference between consonant and dissonant scores rather than plotting consonant and dissonant scores separately.

Experimental design

The research falls within the Aims and Scope of the journal. The research question is well defined and meaningful, and it is stated how the research fills an identified gap.

The methods are generally described in good detail. However, it is important for the reader to understand what chords were used in the stimuli. It is not sufficient to refer to another paper for this information, it should rather be included in the Methods.

One explanation for the lack of consonance perception in the highlanders could be that they did not understand the pleasantness judgement task. How did you ensure that the participants understood the task? Ideally there would be a control experiment where the participants demonstrated the ability to give reliable pleasantness judgements for a different kind of stimulus. Without such a control it is difficult to be confident in the present findings.

Validity of the findings

The participant groups in the study have sizes of six (Italians), six (Nepalese), five (lowland Sherpa), and five (highland Sherpa). As noted by the authors, it is harder to achieve large sample sizes with cross-cultural participant groups. Recruitment difficulties notwithstanding, it is very difficult to be confident in group difference findings when the groups number as low as five participants each, especially since the paper includes analyses at the individual-differences level. In particular, in the mediation analyses we see non-significant effects with nonetheless large point estimates, indicating that the statistical power is too low to identify effect sizes to any reliability. For comparison, the McDermott et al. (2016) study cited by the authors used sample groups of 64 (Study 1) and 50 (Study 2) Tsimane participants. In this sense the present study resembles more of a pilot than a final experiment. Indeed, the authors describe the study as a ‘pilot study’ in the title, but don’t mention the word ‘pilot’ elsewhere in the manuscript, and don’t hold back from describing themselves as being “quite confident about the validity of our conclusion.” If this paper really were documentation of a pilot study, then it would be useful to hear much more about what the authors learned from the piloting experience, and how they would conduct an improved study in the future. As it is, it is unclear that the paper represents an appropriate ‘unit of publication’ for PeerJ.

Additional comments

I reviewed an earlier version of the manuscript for another journal, and made many of the same comments provided above. While the authors did respond to some of the minor points I made in that previous review (e.g. adding ‘pilot’ to the title, fixing some typos, improving some unidiomatic English), they seem not to have responded substantively to any of the major comments. As a result, much of the above text is duplicated from my original review. I hope that this time the authors invest the time to improve the manuscript with the feedback from myself and the other reviewers.

·

Basic reporting

The authors investigated the “consonance effect”, consisting in higher pleasantness judgments for consonant compared to dissonant chords – according to the Western definition of music in Italian and Himalayan participants. They were asked to express pleasantness judgments for consonant and dissonant chords at 1450 m and at 4750 m of altitude, with the aim to evaluate the effect of hypoxia on this task. Moreover, they considered the total mood disturbance as potential mediating factor.
Although I did some works on music, I do not consider myself expert of it, but I have more experience on cognitive processes, exercise and mood also in extreme conditions, so my feedback will be more on these aspects.
In my opinion it is interesting what the authors found in terms of TMD. I expected that between the first and second measurements there would be an increase (and not a decrease) of the total mood disturbance, because of the fatigue/stress/physical efforts (Di Corrado et al., 2014; Dupuy et al., 2014; Murgia et al., 2016). However, I agree with the interpretation of the authors that the concern before climbing vs. the gratification of after it might be an important factor, determining the results they observed.
Although there would be room for a large debate, the study by Shukitt-Hale & Lieberman (1996) – cited in the discussion of the present manuscript – indicates that mood states were adversely affected by duration and level of altitude, but with a main role of physical exertion. In this regard, the study by Murgia et al. (2016) indicates that physical exertion alone (after a multi-stage cycling race, thus independently from the altitude hypoxia) can significantly increase the TMD. Thus, it is possible to speculate that the participants of this study were not particularly affected by physical exertion and that hypoxia per se does not necessarily increase the mood disturbance.
I suggest authors to further work on this part of the discussion, including the observations above.

Minor points
There are some typos/errors to fix (e.g. “evidence” is an uncountable word and its plural is not allowed). I recommend an English proofreading before resubmission.

References
Di Corrado, D., Agostini, T., Bonifazi, M., & Perciavalle, V. (2014). Changes in mood states and salivary cortisol levels following two months of training in elite female water polo players. Molecular Medicine Reports, 9(6), 2441–2446.
Dupuy, O., Lussier, M., Fraser, S., Bherer, L., Audiffren, M., & Bosquet, L. (2014). Effect of overreaching on cognitive performance and related cardiac autonomic control. Scandinavian Journal of Medicine & Science in Sports, 24(1), 234–242. https://doi.org/10.1111/j.1600-0838.2012.01465.x
Murgia, M., Forzini, F., Filho, E., DI Fronso, S., Sors, F., Bertollo, M., & Agostini, T. (2016). How do mood states change in a multi-stage cycling competition? Comparing high and low performers. The Journal of Sports Medicine and Physical Fitness, 56(3), 336–342.

Experimental design

The research question are well defined, relevant & meaningful. The investigation has been conducted rigorously. Methods have be described with sufficient information to be reproducible by another investigator.

Validity of the findings

The data are robust, statistically sound, and controlled. As stated in the Basic reporting, authors should extend the discuss of TMD results.

·

Basic reporting
* * *
For easier-to-read formatting of this review text, please use the Word document
* * *
1. BASIC REPORTING
1.1. Clear and unambiguous, professional English used throughout.
There is redundancy in how the task is described, in the Stimuli And Procedure section:
<STARTQUOTE> participants were instructed that very brief audio clips would be presented one at time, and that they will be required to express a pleasantness judgment
..
The participant was asked to listen to each stimulus and to express a pleasantness judgment
<ENDQUOTE>


At line 126, the same information seems to be repeated unnecessarily, within the span of a few lines:
<STARTQUOTE> an Italian sample was tested both at lower and higher altitudes with the aim to evaluate the possible effect of hypoxia at high altitude. Moreover, a mood questionnaire was also administered to this sample, in order to investigate the effect of hypoxia also on this variable. The second sample was Nepalese, with no or little exposition to Western culture and music. As for the Italian group, also this sample was tested both at lower and higher altitude
<ENDQUOTE>

Line 74:
<STARTQUOTE> the comparison among music preferences expressed by individuals belonging to very different cultures and living in countries very far apart from each other can help in shedding light on this topic
<ENDQUOTE>
This should probably not be a hard logical "and" here: I don’t think that countries need to be "very far apart" from one another to manifest widely different musical cultures/preferences that would be relevant in such a study. This is the case for instance with remote indigenous tribes of the Amazon investigated in previous studies (and perhaps also with the Nepalese communities investigated here, and described in the next passage)‎, that despite being – on the map – relatively close to one another (and close also even to Western-permeated communities of S. America, e.g. to urban Bolivia) are nonetheless very different in their musical traditions.

Finally, the "Aim" section goes into quite a lot of detail about the design and experimental manipulations, detail that is probably better suited (and that is indeed repeated) in the Methods section. The full design is not clear from the "Aim" section, which for instance does not specify whether the Nepalese sample were native of – and were tested at – low or high altitude; this is only clarified later in the Methods.
1.1.1 ~Persistent confusions of terminology
Despite the subtle and complex relationship between the notions of 'pleasantness' and 'consonance' having been explained earlier, and despite this relationship being crucial for this paper, the former term seems to be used inter-changeably with the latter on multiple instances, alongside other several inaccurate usages of these and adjacent terms. I list some of those instances below.

<STARTQUOTE> Some evidences suggested music pleasantness to be a universal principle, independent from the cultural environment: for instance, a preference for consonance was described even in different species,(..)
<ENDQUOTE>
I think this leads to confusion and to a re-ambiguation (instead of a disambiguation) of terms. A subjective feature of music (pleasantness) cannot possibly constitute a universal principle. Furthermore, this wording also seems to ignore the fact that musical pleasantness can in fact arise out of multiple musical features, some not even in the pitch domain, e.g. groove (Witek et al., 2014).

This same confusion continues at line 69:
<STARTQUOTE> (..) suggesting that music pleasantness might not be universal, developing instead with experience and listening habits, namely on culture (McDermott et al., 2016).
<ENDQUOTE>
The cited McDermott paper is about whether or not consonance preference – not musical pleasantness preference – is universal. Again, since these are notions central to the manuscript, I think the effort would be worth making to more clearly use each term in the right context.

The set of inter-changeably used terms includes not just 'pleasantness' and 'consonance' but also the general term 'music preference'. For instance, in the first paragraph of the Discussion, it is stated that
<STARTQUOTE> This first result would exclude the possibility that music preference is innate and biologically determined, because if this hypothesis were true we should have found a similar preference in all the samples tested.
<ENDQUOTE>
Again, I see no reason why consonance preference (which is the only type of preference tested here) would be expressed in such generic terms here, as "music preference". On the other hand, it is warranted to describe musical preference in general (with all of its facets) later in the conclusions, as the authors indeed do:
<STARTQUOTE> we could consider these results as a further evidence in favor of the crucial role of music exposition in music preference
<ENDQUOTE>

Cf line 111:
<STARTQUOTE> there is still a lack of results regarding the effect of altitude hypoxia on music pleasantness.
<ENDQUOTE>
Presumably it is the effect on preferences related to musical pleasantness that is intended here!

Cf line 355:
<STARTQUOTE> our results showed that neither pleasantness, nor consonance-dissonance difference, were influenced by altitude
<ENDQUOTE>
Once again, this is very confusing: isn't the consonance-dissonance difference computed in terms of pleasantness ratings? Then why list this as two findings when it is really just one?

1.2. Literature references, sufficient field background/context provided.
Adequate.

1.3. Professional article structure, figures, tables. Raw data shared.
Could the authors please also include the trial-by-trial data in the raw XLS file, and not just the subject-wise averages? Many thanks.

Figures:
• Fig1
o the figure caption reads "Measurements at low altitude at altitude and high altitude (ML, MA and MAH)". The Methods section only explains two of those: "lower and higher altitudes (respectively ML and MHA)". Presumably MA stands, then, perhaps for middle altitude (?!), however this abbreviation system does not seem to make sense, and is not consistent.
o the x axis contains MHA instead of MAH as in the caption
o it's not explained in the caption or elsewhere what the gray data points represent (e.g. corresponding to the "KJN base")
o the stick figure icon seems to go with triangular data points to indicate measurements in the middle of the time series plot, but not also at the beginning (second data point), where the data point is circular
• Figs 4 and 5
o the values on the y axis are probably better displayed in 0.5 rather than 0.3 increments

1.4. Self-contained with relevant results to hypotheses.
Cf line 54:
<STARTQUOTE> if harmony is the base for the consonance effect, how are consonant and dissonant chords evaluated in those cultures in which harmony is not present?
<ENDQUOTE>
Again, terms are being conflated here. If 'harmony' is to be used, then it would be worth giving at least a short clarification of what this term really taken to mean, in Western music. The sensory dissonance/consonance phenomenon of focus in this study is merely the 'vertical' dimension of harmony (i.e. concerning isolated chord presentations). However, a hierarchically-higher 'horizontal' dimension also exists (not of interest here), where the key notions of voice-leading, counterpoint and diatonic function are brought into the equation. The apparent 'paradox' of people liking dissonance then is simply – and elegantly – explained in this wider framework of harmony, as the simultaneous existence of a purely-negative perceptual connotation, stemming from the vertical/acoustical dimension (essentially: beating and roughness); with a further dimension provided by the horizontal dimension of harmony. This dimension (essentially: Western music conventions) is able to use this dissonance (in addition to other mechanisms) to create and control musical tension.
Relevant references from the music cognition literature, that may well be already known to the authors, should they wish to make this acknowledgement, are (Huron, 2001; Krumhansl, 1990; Terhardt, 1984; Tramo et al., 2001); and perhaps (Aldwell et al., 2010) on the music-theoretical side.


Also, see section 3.2.1 below.

Experimental design

2. EXPERIMENTAL DESIGN
2.1. Original primary research within Aims and Scope of the journal.
Adequate.
2.2. Research question well defined, relevant & meaningful. It is stated how research fills an identified knowledge gap.
2.2.1 ~Hypotheses
<STARTQUOTE> Concerning altitude, due to the absence of previous evidence in this regard, we did not have specific expectations on the effect of hypoxia on music pleasantness evaluation
<ENDQUOTE>
Although too little may be known about the effects of altitude – namely high-altitude-induced hypoxia – on musical preferences to formulate a hypothesis, this factor was nonetheless of sufficient prior interest to investigate and manipulate, thus it would be good if this reasoning were more deeply thought-through. After all, this is presumably what this study aims to bring, over and above similar previous ones such as (McDermott et al., 2016) - see section 4.1 below. For instance, perhaps a link can be made to what prompted this factor to even be considered, as hinted at line 109:
<STARTQUOTE> It is well known that altitude hypoxia negatively affects some aspects of physiology and, specifically, sensory systems (Cingi, Erkan & Rettinger, 2010; Jha, 2012; Ruffini et al., 2015); however, to our knowledge, there is still a lack of results regarding the effect of altitude hypoxia on music pleasantness
<ENDQUOTE>

Although obviously no good in terms of formulating a hypothesis at this stage (which would be mere HARKing), perhaps a slightly more extended discussion starting from what is known about the effect of hypoxia on sensory systems in general would be warranted (perhaps from the wider " Environmentally-modulated metabolic.." project). Currently, this is only briefly touched upon in the Discussion:
<STARTQUOTE> Even if we did not have specific hypotheses in this regard, we also considered altitude hypoxia as a possible influencing variable, because of the well-known affecting role of such condition in sensory systems and cognitive performance (Cingi et al., 2010; Jha, 2012; Limmer & Platen, 2018; Ruffini et al., 2015), but our results showed that (..).
<ENDQUOTE>
2.3. Rigorous investigation performed to a high technical & ethical standard.
Unfortunately, there are multiple issues here - see subsections below.
2.3.1 ~Design/factors
Among the limitations typical of field studies in extreme environments, the authors cite: the difficulty in testing participants belonging to remote cultures, testing having been done at different altitudes, and the absence of control tasks. It would however be good to acknowledge that a further limitation of the study (whether one typical of such field studies, or simply a limitation of this specific study) is that the full factorial design was not tested, for instance that the low-altitude Sherpas were not also tested at high altitude (and vice-versa).

In the same vein, something that also strikes the reader as an arbitrary element of "missing data", is the fact that only the Italian group was measured on the Profile Of Mood State (POMS), with subsequent analyses of the effect of altitude on TMD; whereas it seems to have made sense equally to collect this data also for the other two native-low-altitude groups (She_L and Nep_L - see table below)! As suggested elsewhere, perhaps this can also be explained as a limitation typical of expedition/trekking studies.

The (incomplete) mixed design employed might be more clearly laid out in a table that clarifies the relevant conditions, manipulations and factor. This table might look something like:
Western music familiar
yes no
Altitude L She_L
Ita_L Nep_L
H Ita_L! Nep_L!
She_H
where:
- _L or _H denotes that the group's native altitude is low or high respectively;
- "!" refers to testing done in conditions of altitude different to any one group's native altitude.

Clarifying the design in this way would more clearly highlight which comparisons were within subjects (i.e. effect of altitude-change from low to high) vs between subjects (i.e. effect of low vs high native altitude), even if the effect of the Western factor (unlike Altitude) is in fact tested not via factorial ANOVA but via mediation analysis.

The providedTable 1 does not clarify these essential design parameters. Also, this table is, to my eyes, not necessarily suitable for the main article text, since the full raw-data table (with extra data on ratings) is available, and since variables such as "foreign language spoken" are not at all essential to the study. This Table also has some missing data (e.g. in the Y/Mus column).

I find it odd that, although She_L and She_H are two different groups, the authors chose to describe their overall sample as consisting of three "groups": Italians, Nepalese and Sherpa, with 2 "subgroups" in the latter group, on native altitude grounds. Perhaps clarify that both of the two 'subgroups' and the two 'groups' are in fact on the same hierarchical level, and together form the levels of your Ethnicity factor (as used in some of the ANOVAs).


Cf line 130:
<STARTQUOTE> As for the Italian group, also this sample was tested both at lower and higher altitude, in order to evaluate the possible interaction between culture and hypoxia on the consonance effect.
<ENDQUOTE>
I think it would be good to clarify whether hypoxia is a temporary 'acute' state (that low-altitude-native populations are susceptible to), or a 'chronic' state (typical of high-altitude-native populations).
As I understand (e.g. from here), hypoxia refers by definition to the former, i.e. the effect of high altitude upon an individual NOT habituated to high altitudes. If that’s true, then comparing the performance of a low-altitude-native group at low and at high altitude is not qualitatively the same as comparing a low- vs a high-altitude-native group, as hypoxia (as construed above) only affects the low-altitude natives. Thus, simply distinguishing the statistical nature of each comparison (Altitude being a within- and between subjects factor, respectively), I feel it is a problem that the NativeAltitude factor (yes/no) is nowhere modelled in this.


A further unrelated (and major) problem is that, according to the XLS file provided, the SherpaLow and SherpaHigh groups seem to not be homogeneous in terms of being familiar/not with W. music (one participant who gave the opposite response, in each group); and the same for the Nepalese group.
Despite this, the "familiarity with Western music" factor seems to be defined and tested categorically, along precisely the lines of group membership, as if this variable was homogeneous within each group. This is even more of a problem given the very low sample size (N=5 for She_L).
Thus, the two categorical factors are haplessly confounded, and only the first Mediation Analysis, which uses the binary variable of familiarity with Western music, with no group attribution is reliable! However, that analysis seems to not be considered central by the authors (Fig3 only shows the second Mediation Analysis), and instead the main conclusion that consonance preference is cultural, is primarily based on the ANOVAs, where the Western factor is, as I explain, imperfectly coterminous with group membership as used as a factor.

Related, while the difficulty of finding participants in remote groups can perhaps explain the non-homogeneity (in terms of familiarity with W. music) in the Sherpa group, perhaps the authors can explain why this was also the case in the Nepalese group, where presumably (?) those difficulties did not apply.
2.3.2 ~Analyses
<STARTQUOTE> Then, by means of two mixed repeated-measures ANOVAs, Ethnicity (Italian, Nepalese, Sherpa) was considered as a between-subjects factor, and Chord (consonant, dissonant) was used as a within-subject factor. Pleasantness evaluations recorded at the lower altitude were used as the dependent variable in the first ANOVA, and those recorded at the higher altitude were used in the second ANOVA.
<ENDQUOTE>
This ANOVA would be described as a mixed-model ANOVA, as opposed to a "mixed repeated-measures" ANOVA, as it has both a within- and a between-subjects factor.

<STARTQUOTE> In all ANOVAs post-hoc tests were carried out by means of Duncan test.
<ENDQUOTE>
Does this multiple-comparisons-correction measure also correct for the fact that – due to the incompleteness of the native altitude X tested altitude X ethnicity X Western music familiarity design matrix (see above) – you have had to use four separate ANOVAs? If not, I would suggest to mention this as a statistical limitation, in the same place where the low number of subjects is mentioned; and explain perhaps that both were consequences of the difficulty of getting (enough) subjects in each cell of the design.
The low sample size limitation might also be linked/qualified in terms of the word "pilot" that appears only in the title but nowhere discussed/qualified in the paper.
2.3.3 ~Stimuli
It would be good if the 24 chords were made a part of the Supplementary Materials – in both music notation as well as waveform/MP3 format. Otherwise, it is very difficult for readers to imagine what those chords sounded like, in order to judge the amount of dissonance in each for themselves. Prete et al. 2015's SM does not seem to hold this information either.

Also, the number of unique chord types (that is, disregarding transpositions) used as exemplars here across the consonant and dissonant categories, was N=6. This makes for little variability within each category, inasmuch as this variability is used for inference (generalisation). This is at least the case in comparison to the variability found in other similar studies, i.e. behavioural studies, based on single-chord presentations (chords of 3 notes or more), with definitions of dissonance not based on harmonicity.
Such studies used stimulus sets with e.g. N=15 chords (Lahdelma & Eerola, 2016), N=28 chords (Park et al., 2011), N=80 chords (Popescu et al., 2019), etc. Outliers in this sense can be found in the literature, e.g. N=3 stimuli (Foo et al., 2016), although these are generally fMRI studies, with the corresponding limitations.
Still, this low variability should, in my opinion, be acknowledged as a limitation, and perhaps be again linked to the limitations that existed upon the total testing time.
2.4. Methods described with sufficient detail & information to replicate.
Adequate.

Validity of the findings

3. VALIDITY OF THE FINDINGS
3.1. Meaningful replication encouraged where rationale & benefit to literature is clearly stated.
The hypoxia hypothesis not being clearly formulated (see section 3.2.1) is particularly problematic since the study of the effect of hypoxia seems to be this study's main addition, above and beyond McDermott 2016, as indeed stated in the Discussion at line 384:
<STARTQUOTE> Compared to [McDermott's (2016)] findings, the present study adds the evidence of null effects of other demographic information on music preference, as well as it shows a null effect of hypobaric hypoxia on the pleasantness of music.
<ENDQUOTE>

3.2. All underlying data have been provided; they are robust, statistically sound, & controlled.
<STARTQUOTE> the difference of pleasantness evaluation between consonant and dissonant chords was also calculated, obtaining an overall measure in which negative scores corresponded to unpleasant judgments, positive scores corresponded to pleasant judgments and a score of 0 corresponded to a neutral judgment (neither pleasant nor unpleasant).
<ENDQUOTE>
I find this description utterly confusing, and not at all matching with the fact that each pleasantness judgment was rated on a 4 point-Likert scale. I expected to read that negative scores corresponded to a dissonance preference, and positive to a consonance preference, but this is not what this text states!
Furthermore, this description confusingly suggests that the consonant/dissonant manipulation was done within-chord (manipulations upon a single starting/baseline chord), whereas that is not how the Stimuli section describes them – these were just different chords. Thus, I don’t see how average pleasantness ratings that are equal between Cons and Diss chords amount to a "neutral judgement", when at most this can be described as equal preferences (e.g. the subject likes both Cons and Diss chords).
3.3. Conclusions are well stated, linked to original research question & limited to supporting results.
The statistics are reported thoroughly and completely. However, as per my observations for the incompleteness of the factorial design, several aspects of the four ANOVAs strike me as odd:
- The first two ANOVAs use partly overlapping groups (Ita, Nep), whereas the Sherpa groups are different between the first and second ANOVAs. The statistics do not "know" (i.e. model) this, and so the ANOVA assumptions regarding the variance within each population are not met. Thus, I do not think this kind of analysis allows drawing results that pool from across this dataset, but instead imposes they be treated as independent results.
- Fig 4 is meant to show the 3-way ANOVA (with altitude included as a factor), but only the Ethnicity and Chord factors are shown in the bar plot. Data should also be broken down by Altitude, even if there is no sign. effect of it.
- The result of the first two ANOVAs are nowhere plotted; perhaps these could at least go into the SupplMat?
- You say:
<STARTQUOTE> Furthermore, consonant chords were judged as more pleasant by lowlanders than by highlanders
<ENDQUOTE>
But I thought the relevant DV was meant to be the difference, upon which what you call the "consonance effect" is based; thus, reporting this effect on the raw Consonance DV in isolation seems to me rather opportunistic, or at best a posthoc side-track.


Also, the second mediation analysis (Fig 3) used both recorded variables (Consonance, Dissonance) as well as their difference, as dependent variables; whereas the significant effect in the first mediation analysis is reported only for the Difference measure – although presumably once again all three were tried as DVs?...


The first paragraph of the Discussion states
<STARTQUOTE> the effect was completely absent in highlander Sherpas, the subsample who declared to listen to music less frequently and, importantly, not to listen to Western music. This latter result is the most important of the present study, because it shows that music exposition is the key factor determining the consonance effect:
<ENDQUOTE>
Whereas in most of the paper, control for enculturation as a possible cause of consonance preference is done by the variable of exposure to Western music, now the authors unexpectedly also throw in the variable of amount of music (of any kind, not just Western) listened to, the "Music/day (hours)" variable in the XLS.
The reasoning behind mixing notions like this is to me, again, incomprehensible: although it can be assumed that any non-Western music remaining in those listening habits is probably not consonance-heavy, there is no assurance of this at all. But even if this assumption is correct, then surely even frequent listening of this type of music should still lead to an absent consonance effect, right? And if this argument is correct, then likewise low values of music listening (as the highlander Sherpas report) should not be taken as proof that consonance preference is culturally-based; only their reported low familiarity with Western music should be taken as such proof - but there, we have the issue presented above with non-homogeneity in the She_H and Nep groups in terms of this variable.


<STARTQUOTE> In the present study we also explored the possible effect of some demographic features on the consonance effect, such as age, schooling and playing music: the results revealed that none of these variables influences music pleasantness, stressing that the exposure to a particular kind of music is the only variable responsible for the consonance effect.
<ENDQUOTE>
Since not ALL demographic variables were tested, but only a select few, I think the conclusion needs to be toned down, by qualifying the term "only", e.g. to say that exposure is likely the most salient predictor.


<STARTQUOTE> Only the mood was influenced by hypoxia, with the Italian sample revealing lower mood disturbance scores (index of tension, depression, anger, fatigue and confusion) at higher compared to lower altitudes. We speculate that this difference could be due to the fact that once climbed the mountain, the more positive mood can be due to a kind of gratification compared to the possible concern before the climb. This result agrees with the findings of Karinen and Tuomisto, who reported that during a prolonged expedition in Everest mountain well-motivated participants were capable to maintain a good mood state (Karinen & Tuomisto, 2017); on the contrary, Stavrou and colleagues, who reported that hypoxia was capable to negatively affect the mood, exacerbating the effect of bed rest and ambulatory confinement (Stavrou et al., 2018). Other evidences linked mood and altitude hypoxia, affirming that mood states were adversely affected by duration and level of altitude, but with a main role of physical exertion (Shukitt-Hale & Lieberman, 1996). Heinrich and colleagues recently affirmed that acclimatization plays a big role in cognitive and mood alterations during a sojourn at altitude (Heinrich et al., 2019). To be noted that none of our participants suffered remarkable altitude sickness symptoms. Thus, from these findings and from our results, we support the idea that hypoxia may increase the mood disturbance during non-rewarding conditions or in association with altitude sickness symptoms, but that hypoxia per se, or hypoxia during motivating conditions does not promote mood disturbances.
<ENDQUOTE>
I fail to see how this unexpected – and extended – discussion of the effects of altitude on mood is relevant in the Discussion of a paper on the universality of consonance preference, especially when the argument does not come back to this focal topic.


<STARTQUOTE> the fact that our findings strictly resemble those recently described by McDermott and colleagues allows us to be quite confident about the validity of our conclusion
<ENDQUOTE>
The conclusion of the present study indeed rest heavily on McDermott's 2016 study, in supporting the conclusion that consonance preference is cultural and not biological. To note, however, is that said McDermott study has been heavily criticised methodologically, e.g. by Bowling et al. (2017), Zatorre (2016) and others. This might or might not convince the authors to reconsider the degree to which their results (themselves qualified, for what it's worth, by the criticisms in this review), together with the McDermott result, really constitute strong evidence for the stated hypothesis. Might a more ample/general framework be more suited, e.g. Cross' proposal of music as a “biocultural phenomenon” (2003), at least until the literature has some more definitive results on the biology/culture question?


<STARTQUOTE> in a cross-cultural study with Western and African (Cameroon) participants, Fritz and colleagues (Fritz et al., 2009) found no group differences in judging both emotional content (happy, sad, fearful) and pleasantness of Western and African music, suggesting a biological basis of music processing, independent from the culture.
<ENDQUOTE>
It seems to me the authors only incompletely acknowledge the arguments in favour of the opposite view. Stated as above, the reader might remain unconvinced as to the side taken by the authors. This is because the Fritz et al. study actually concluded that "sensory dissonance [seems to] universally influence the perceived pleasantness of music". Thus, if this study is cited as proof for a biological basis of music pleasantness, it seems that, for authors to retain a conviction in the cultural basis hypothesis, some further discussion is necessary!

Additional comments

4. GENERAL COMMENTS
* * *
For easier-to-read formatting of this review text, please use the Word document
* * *
In general, I found the premise of the study justified. The highlighted need for cross-cultural studies, and the attempt to accrue more evidence to further address the nature-nurture question in music cognition (with regards to consonance preference, and perhaps more generally), are all valid. As a behavioural study on consonance preference across cultures, what this study – part of a wider empirical project to study aspects of human physiology across various conditions – is certainly a very welcome addition to the literature.

However, as you will read in the sections below, I am afraid I also found the manuscript unsatisfactory in several crucial respects, to do with the methodology, hypotheses, clarity of writing, and coherence of reasoning. I do think that at least the most substantive of my comments would have to somehow be addressed before considering publication, which I hope that the Authors will consider doing (and hope that the Editor will allow the revision). My recommendation is essentially "accept conditional upon satisfactory major revisions".

I welcome further discussion and would be happy to review a revised version of the manuscript. If the paper is eventually accepted (which I hope it is, assuming satisfactory revisions), then I would very much hope and encourage the authors to agree to publish the review history, in order to continue (what I regard to be) the good policy of open peer review practiced by PeerJ (and other journals).

I have largely structured my comments under the suggested headings of the PeerJ review template. I have included some subheadings of my own to those, their names prefixed with a ~. When my comments pertain to a particular section of the text, I am quoting that section using
<STARTQUOTE> text formatted as such.
<ENDQUOTE>
I have tried to place comments inside the relevant subheadings as explained, rather than structuring them in terms of major/minor comments – hopefully it will be clear (also from this heading structure) which comments relate to 'deep structure' and which to 'form'. Please also note the annotations made in the PDF version of the manuscript, which contain predominantly minor/surface comments (e.g. suggestions re clarity of phrasing/wording etc.), that the authors may wish to also take into account.
* * *
5. REFERENCES
Aldwell, E., Schachter, C., & Cadwallader, A. (2010). Harmony and Voice Leading. Cengage Learning.
Bowling, D. L., Hoeschele, M., Kamraan, Z. G., & Fitch, W. T. (2017). The Nature and Nurture of Musical Consonance. Music Perception, 35(1), 118–121.
Cingi, C., Erkan, A. N., & Rettinger, G. (2010). Ear, nose, and throat effects of high altitude. European Archives of Oto-Rhino-Laryngology: Official Journal of the European Federation of Oto-Rhino-Laryngological Societies (EUFOS): Affiliated with the German Society for Oto-Rhino-Laryngology - Head and Neck Surgery, 267(3), 467–471. https://doi.org/10.1007/s00405-009-1016-6
Cross, I. (2003). Music as a Biocultural Phenomenon. Annals of the New York Academy of Sciences, 999(1), 106–111. https://doi.org/10.1196/annals.1284.010
Foo, F., King-Stephens, D., Weber, P., Laxer, K., Parvizi, J., & Knight, R. T. (2016). Differential Processing of Consonance and Dissonance within the Human Superior Temporal Gyrus. Frontiers in Human Neuroscience, 10. https://doi.org/10.3389/fnhum.2016.00154
Huron, D. (2001). Tone and Voice: A Derivation of the Rules of Voice-Leading from Perceptual Principles. Music Perception, 19(1), 1–64. https://doi.org/10.1525/mp.2001.19.1.1
Jha, K. N. (2012). High Altitude and the Eye. Asia-Pacific Journal of Ophthalmology (Philadelphia, Pa.), 1(3), 166–169. https://doi.org/10.1097/APO.0b013e318253004e
Krumhansl, C. L. (1990). Cognitive foundations of musical pitch. Oxford University Press, USA.
Lahdelma, I., & Eerola, T. (2016). Mild Dissonance Preferred Over Consonance in Single Chord Perception. I-Perception, 7(3), 2041669516655812. https://doi.org/10.1177/2041669516655812
Limmer, M., & Platen, P. (2018). The influence of hypoxia and prolonged exercise on attentional performance at high and extreme altitudes: A pilot study. PLOS ONE, 13(10), e0205285. https://doi.org/10.1371/journal.pone.0205285
McDermott, J. H., Schultz, A. F., Undurraga, E. A., & Godoy, R. A. (2016). Indifference to dissonance in native Amazonians reveals cultural variation in music perception. Nature, 535(7613), 547–550. https://doi.org/10.1038/nature18635
Park, J. Y., Park, H., Kim, J., & Park, H.-J. (2011). Consonant chords stimulate higher EEG gamma activity than dissonant chords. Neuroscience Letters, 488(1), 101–105. https://doi.org/10.1016/j.neulet.2010.11.011
Popescu, T., Neuser, M. P., Neuwirth, M., Bravo, F., Mende, W., Boneh, O., Moss, F. C., & Rohrmeier, M. (2019). The pleasantness of sensory dissonance is mediated by musical style and expertise. Scientific Reports, 9(1), 1070. https://doi.org/10.1038/s41598-018-35873-8
Ruffini, R., Di Giulio, C., Verratti, V., Pokorski, M., Fanò-Illic, G., & Mazzatenta, A. (2015). Adaptation of olfactory threshold at high altitude. Advances in Experimental Medicine and Biology, 837, 19–22. https://doi.org/10.1007/5584_2014_70
Terhardt, E. (1984). The Concept of Musical Consonance: A Link Between Music and Psychoacoustics. Music Perception, 1(3). http://search.proquest.com/docview/1300644337/citation/33847C4B99D341EAPQ/1
Tramo, M. J., Cariani, P. A., Delgutte, B., & Braida, L. D. (2001). Neurobiological foundations for the theory of harmony in western tonal music. Annals of the New York Academy of Sciences, 930(1), 92–116.
Witek, M. A. G., Clarke, E. F., Wallentin, M., Kringelbach, M. L., & Vuust, P. (2014). Syncopation, Body-Movement and Pleasure in Groove Music. PLoS ONE, 9(4), e94446. https://doi.org/10.1371/journal.pone.0094446
Zatorre, R. (2016). Human perception: Amazon music. Nature, 535(7613), 496–497. https://doi.org/10.1038/nature18913

---

## Round 0.2 · Minor Revisions

Two reviewers are happy with your revision but one requires some further amendments and clarifications. Please address the Reviewer's comments and I would be happy to reconsider your paper.

Reviewer 1 ·

Basic reporting

This has been significantly improved since the last submission.

Experimental design

The methodological reporting has been improved, with useful information about the musical stimuli. One note is that the authors included the chord <C, E, A flat> in the 'consonant' stimulus set, but in Western music this chord (termed the augmented triad) is generally considered dissonant and inharmonic. This is not a huge problem for the validity of the finding, but in future studies the authors might consider instead using the consonance-dissonance stimuli of Josh McDermott's group, which fulfil a similar function to these stimuli but are more consistent with consonance judgements in Western music.

Validity of the findings

The authors have improved their commentary on their findings, making a better job of restricting their conclusions according to the methods employed and the results obtained.

Additional comments

The paper has been improved a lot, and now the main weakness is in the Introduction. Still, as commented by me and by Tudor Popescu, the theoretical account of consonance is somewhat incoherent. The paper really needs to provide a clear definition of consonance, and it needs to explain whether consonance is being treated here as being terminologically equivalent with pleasantness, as in the operationalisation of Josh McDermott's papers. I dislike the term 'consonance effect' - I have not seen it before literature, and it is problematic because it implies that the only valid form of consonance is the consonance system of Western music.

The definition of harmonicity is likewise incoherent. The authors write this:

> A harmonic chord (a set of notes played together) sounds as pleasant if it is consonant, and consonance is defined by the specific intervals among notes (Bowling & Purves, 2015). According to this view, in fact, consonance is associated with pleasantness, and dissonance with unpleasantness (the so-called “consonance effect”; Prete et al., 2015, 2019).

Here, it just seems like you're equating harmonicity with consonance and pleasantness. This is not useful. Here is how I think the terminology should be defined (not in these specific words, of course):

Consonance - a concept in Western music theory concerning how notes may be combined into chords. Consonance is associated with pleasantness, stability, and relaxation.

Pleasantness - this is how you operationalise consonance in the present study, consistent with many previous studies (e.g. Bowling et al., McDermott et al.) While pleasantness is only a part of the music-theoretic notion of consonance, it is the easiest to study with nonmusicians and nonWesterners.

Harmonicity - this is when the pitches in the chord are related by simple integer ratios of frequencies. It is a potential contributor to pleasantness and consonance, with this effect being possibly moderated by cultural exposure.

Interference - this is when the spectral components in a chord are close enough to elicit masking and beating. It is a potential contributor to pleasantness and consonance, with this effect being possibly moderated by cultural exposure.

I hope this is clear?

Minor comments:

l. 60: 'deepen' should be 'deeper'

l. 104-106: unclear, please clarify

l. 124: not sure what you mean by lacking modulation frequency

l. 125-126: the definition of tonal hierarchy is not very accurate, please revise (you can have a tonal hierarchy without notes deviating from the scale)

l. 252: pc should be PC

l. 411: complains should be complaints

·

Basic reporting

Fine.

Experimental design

Fine.

Validity of the findings

Fine.

Additional comments

The authors have satisfactorily fulfilled all my requests.

·

Basic reporting

Adequate.

Experimental design

Adequate.

Validity of the findings

Adequate.

Additional comments

I congratulate the authors for having thoroughly addressed my concerns, comments and suggestions. The text has been much improved and clarified, and so have the figures. I regard this revision as being suitable for publication.
As a small remaining point, I would still suggest to include the music notation (score) for the 24 chords, if available; I did not see this in the Supplemental Materials. If this is not available, the fact that at least the audio files are now included is certainly still helpful.

---

## Round 0.3 · accepted · Accept

It seems to me that you have adequately addressed all the issues raised by the Reviewers and I am therefore happy to accept the paper for publications.